# Living with Pain and Looking for a Safe Environment: A Qualitative Study among Nursing Students with Dysmenorrhea

**DOI:** 10.3390/ijerph17186670

**Published:** 2020-09-13

**Authors:** Elia Fernández-Martínez, Ana Abreu-Sánchez, Jorge Pérez-Corrales, Javier Ruiz-Castillo, Juan Francisco Velarde-García, Domingo Palacios-Ceña

**Affiliations:** 1Department of Nursing, University of Huelva, Avenida Tres de Marzo s/n, 21071 Huelva, Spain; elia.fernandez@denf.uhu.es (E.F.-M.); abreu@denf.uhu.es (A.A.-S.); javirucastillo@gmail.com (J.R.-C.); 2Department of Physical Therapy, Occupational Therapy, Physical Medicine and Rehabilitation, Research Group of Humanities and Qualitative Research in Health Science of Universidad Rey Juan Carlos (Hum&QRinHS), Avenida Atenas s/n, 28922 Alcorcón, Spain; domingo.palacios@urjc.es; 3Department of Nursing, Red Cross College, Instituto de Investigación Sanitaria Gregorio Marañón (IiSGM), Universidad Autónoma de Madrid, Calle Reina Victoria 28, 28003 Madrid, Spain; jvg@cruzroja.es

**Keywords:** dysmenorrhea, pelvic pain, nursing students, qualitative research

## Abstract

Dysmenorrhea refers to chronic pain associated with menstruation that is often accompanied by other symptoms. Primary dysmenorrhea (PD) occurs without any associated pelvic disease. Nonetheless, it may negatively affect women’s quality of life. Among university students, dysmenorrhea decreases academic performance and is a cause of absenteeism. The purposes of our study were to describe how nursing students experienced PD and the changes affecting their body and mood. A qualitative case study was performed among 33 nursing students with PD. Data were collected through five focus groups (with two sessions each) and 10 researchers’ field notes. We used a video meeting platform to conduct the focus groups. A thematic analysis was performed, and the Standards for Reporting Qualitative Research and the Consolidated Criteria for Reporting Qualitative Research guidelines were followed. Three main themes emerged from the data: (a) living with dysmenorrhea, with two subthemes: menstruation and pain; (b) body changes and mood swings; and (c) seeking a safe environment, with three subthemes: safe environment, unsafe environment, and key safety aspects. Students considered menstruation to be negative and limiting, causing physical and mood changes, making them feel less attractive, and conditioning their way of dressing and relating.

## 1. Introduction

Dysmenorrhea is the most common gynecological disorder worldwide among women of childbearing age [1,2]. Worldwide, the estimated prevalence is 66.6–75.2%. [1]. Concretely, in Spain, it is estimated to affect 56–62% of the female population [3,4] and around 75% of university students [5,6]. Dysmenorrhea is defined as chronic spasmodic pain that occurs immediately before and/or during menstruation, located in the pelvic and/or lower abdominal region, that can last for hours or several days. It is accompanied by other symptoms such as nausea, vomiting, dizziness, headache, irritability, and depressive symptoms [7,8,9]. Two types of dysmenorrhea are identified: primary dysmenorrhea (PD) and secondary dysmenorrhea (SD). In the case of PD, this occurs without any associated pelvic pathology. Some authors have related PD to psychogenic aspects [10]; however, it is currently associated with biochemical alterations, such as an excess secretion of prostaglandins and vasopressin, in a complex context involving the immune, endocrine, and vascular systems [2,10,11,12]. The most commonly used treatment is non-steroidal analgesics and anovulatory drugs; also, self-medication is common [5,13,14]. In addition, SD is related to the presence of a pelvic pathology that causes the symptoms, most often endometriosis. Although SD usually responds to the same treatment as PD, in some patients, surgery may be necessary [15,16]. Previous studies have identified a great sociocultural influence in relation to all aspects of menstruation, which has traditionally been a taboo subject around which there are numerous myths in different cultures [17,18]. These aspects must be taken into consideration given their potential influence on sexual health, since, for example, the concealment of menstrual aspects has a negative impact on the correct assessment and treatment of problems by health professionals [19].

Both types of dysmenorrhea negatively affect a woman’s quality of life [20,21,22]. In the case of university students, there are further connotations, as dysmenorrhea impacts academic performance and has been related to decreased concentration and menstrual migraine, among other symptoms, entailing greater absenteeism and presenteeism during menstruation [1,20,23,24]. This situation is made worse by the fact that many young women do not consult health professionals and delay seeking care because of embarrassment, distrust of professionals, or acceptance of the presence of pain as something inevitable [5,23].

At present, there are few international studies [25,26,27] and none in Spain, to describe and analyze the perspective and experience of young female students suffering from dysmenorrhea and how it influences their daily lives. Previous studies carried out in different countries agree that a large number of women of different ages normalize their condition and do not seek health care [14,28,29], which contrasts with other studies highlighting that dysmenorrhea negatively affects women’s quality of life; in the case of university students, it has an impact on academic performance and is related to absenteeism and presenteeism [1,23]. The majority of nursing students are young women, and therefore as such, they potentially experience menstrual pain. Absenteeism due to dysmenorrhea may be more of a concern for these students because their training requires a minimum of 2300 h of clinical practice, according to European Directives, for the acquisition of their professional skills [30]. On the other hand, rigid regulations regarding attendance and control of absenteeism can increase presenteeism, which is understood as the practice of attending work or clinical training when a person is ill and not fully functioning [23,31]. Furthermore, it has been shown that in the nursing context, this practice increases the risk of medication errors and may decrease patient safety [32]. Therefore, there is a need to deepen the experience of nursing students with primary dysmenorrhea, which is likely to influence their perspective on caring for their future patients when they graduate. In addition, previous studies [25,26,27] have reported that dysmenorrhea is not considered a problem for health care providers and society, although dysmenorrhea limits women’s daily life. The experience of these women needs to be described and analyzed to help implement care that addresses the perceived needs of women with dysmenorrhea, with the aim of providing comprehensive care from a gender perspective [33] and a constructivist framework [34].

The objectives of our study were to describe the following: (a) how nursing students experienced PD, (b) how students experienced the associated body and mood changes, and (c) the impact of pain in their lives. The question that guided our study was: What is the perspective of women with PD?

## 2. Materials and Methods

The guidelines for conducting qualitative studies established by the Consolidated Criteria for Reporting Qualitative Research (COREQ) [35] were followed, together with the Standards for Reporting Qualitative Research (SRQR) [36].

### 2.1. Design

A qualitative descriptive case study with a holistic single-case design was conducted [34,37,38] using focus groups (FGs). Qualitative methods are useful for understanding the beliefs, values, and motivations that underlie individual health behaviors [39]. A case study is a research design that examines a specific phenomenon in a real-life setting, and it can be used to describe participants’ experiences regarding care or diseases [38]. A case study may be formed of different units, which help to describe a phenomenon. These units may be different participants who are connected by the phenomenon under study [34,37,38]. In this study, the phenomenon under study is the impact of PD among nursing students. In addition, FGs were used to study unexplored situations or those that were difficult to access via other methods [34,40], as well as confirming the hypothesis, developing questionnaires, and designing intervention programs [40]. FGs have been used for the study of female pelvic pain and dysmenorrhea [41] and to understand the challenges and experiences related to menstruation in girls and adolescents from rural communities of the Colombian Pacific region [42].

### 2.2. Research Team

Six researchers (four men and two women) participated in this study, three of whom had experience in qualitative study designs (DPC, JPC, JFVG). Four had PhDs in health sciences and were not involved in clinical activity. Prior to the study, the positioning of the researchers was established regarding the theoretical framework, the researchers’ beliefs, their prior experience, and their motivation for the research [35]. Researchers based their approach on a constructivist paradigm [39]. This paradigm was based on the assumption that human beings construct their own social reality, and that knowledge is built through increasingly nuanced reconstructions of individual or group experiences [37]. In constructivism, individuals develop meanings of their experiences. These meanings are varied and multiple, leading the researcher to seek a complexity of perspectives. The goal of the research is to rely on the participants’ views of the situation [39]. The main belief was that young women living with PD must learn to bear with these symptoms, and they integrate their management of this condition into all aspects of their life. 

### 2.3. Participants

Purposeful sampling was performed by selecting cases or units (individuals, groups of individuals, institutions) based on their capacity to provide relevant information in response to the research questions [43]. The inclusion criteria consisted of the following: (a) nursing students who were studying at the University of Huelva (https://enfe.acentoweb.com/) during the study period, (b) PD: experiencing menstrual pain for which no underlying cause had been identified [28], (c) suffering from menstrual pain at least once in the last six months [24,25,44], during at least three periods per year [3], and (d) with moderate-severe pain intensity using the visual analog scale (VAS equal to or greater than 4 out of 10) [5,45,46].

The researchers offered all female nursing students from the four Degrees of Nursing at the University of Huelva the possibility of voluntarily participating. Participation in the study was unrelated to any subject or academic activity/work. Out of 403 students in total, 220 met the inclusion criteria, and 43 agreed to participate voluntarily. The first contact was established with the professor of nursing who placed the researchers in touch with the potential participants. During a second face-to-face meeting, the study was explained to the nursing students. Finally, two weeks later, a third meeting took place during which participation was confirmed and informed consent was obtained. Thirty-three female nursing students participated in the study. The mean age of the participants was 22.72 (SD 3.46) years. There were no dropouts.

### 2.4. Data Collection

In order to examine different perspectives within the same group, FGs were held to acquire an understanding of the problems faced by the group and to aid the identification of values and norms [43]. This method of data collection is congruent with the design of the case study [38]. Focus groups generally consist of 6–12 participants [43]. In addition, smaller focus groups give the participants more time to voice their views and provide more detailed information, while participants in larger focus groups may a generate greater variety of information [43].

The FGs were constructed on the basis of the criteria of homogeneity, comprising participants studying at the University of Huelva (nursing students). In order to assign participants to each FG, the students were randomly distributed, maintaining the same proportion of participants in order to ensure uniform numbers in all the FGs. Randomization was performed in order to avoid a selection bias within the nursing students groups. Five FGs were formed, without exceeding the limit of 10, as a higher number would not provide further relevant information [47].

The FGs followed a uniform structure [48]. Each FG comprised between 6 and 9 participants, as group sizes below 4 can make it difficult to sustain a meaningful discussion and a group of >10 may prove difficult to manage [35]. See Table 1.

### 2.5. Focus Group Procedure

The FGs were conducted using a digital platform (https://zoom.us/). This platform enabled the simultaneous participation of groups of people who received an invitation by the members of the research team, and it also enabled audio and/or video recording [48]. Each participant had the option of participating in the FGs with or without activating their video camera, allowing them to choose whether to be recorded in audio or video. At the beginning of the FGs, each participant identified themselves upon entering the platform, so that all participants could be recognized while participating.

The FGs were conducted by a moderator and an observer [49]. The moderator posed questions to which each participant responded, speaking in turns. To request participation, participants raised their hands in the application chat box and the moderator assigned the order of participation. Thereafter, the moderator posed further questions, based on the issues that were brought up in the discussion, in order to further explore or clarify aspects, either individually or with the whole group [49]. The observer supported the moderator, identifying key points and taking field notes (Table 2). In order to comment on and report on the participants’ gestural communication and/or new areas of interest to be investigated, the observer privately messaged the moderator via the corresponding chat box. 

The FGs were conducted in Spanish. A question guide was used [27], which was focused enough in order to gather information on the area of study, although it was open enough to stimulate discussion and interaction among the participants [43]. See Table 3.

All the FGs were audio and video recorded. Permission for recording the FGs was sought prior to performing the recordings. The total duration of the recordings was 522 min, with a mean duration of 52.2 min.

Data collection was pursued until the researcher achieved information redundancy, at which point no new information emerged from data analysis (in our study, this occurred in FG5) [43].

### 2.6. Data Analysis

Three researchers with experience in qualitative studies (JFVG, JPC, DPC) performed the analysis of the FG data. First, an analysis of each FG was performed. Afterwards, the results of the initial analysis were subsequently merged in joint sessions, during which the data collection and analysis procedures were discussed. In the case of differences of opinion, theme identification was decided by consensus.

Data collection was based on a full verbatim transcript of the FGs and the researcher field notes [43,50]. A thematic analysis [50] approach was used on the data, involving an initial descriptive analysis of the transcribed texts (words, sentences, and metaphors directly from the text). Then, the data collected were reduced to codes in order to identify emerging topics. Subsequently, codes were clustered into categories in order to define the main topics. The final outcome was the identification of themes emerging from the data. No qualitative software was used on the data.

### 2.7. Quality Criteria

The criteria for guaranteeing trustworthiness as cited by Guba and Lincoln were followed [37]. The techniques performed and the application procedures used to control trustworthiness are described in Table 4.

### 2.8. Ethical Considerations

This study was approved by the Research Ethics Committee of Huelva (Code: 9/19) and conducted in accordance with the principles articulated in the WMA Declaration of Helsinki (Ethical Principles for Medical Research Involving Human Subjects). All participants provided written informed consent prior to participating in this study.

## 3. Results

The study duration was from 1 January 2020 to 1 June 2020. Three themes were identified: (a) living with dysmenorrhea, with two subtopics, menstruation and pain; (b) body changes and mood swings, and (c) looking for a safe environment, with three subthemes; safe environment, unsafe environment, and key safety aspects. In order to facilitate the traceability and identification of the results obtained, these are accompanied below by excerpts of transcripts. (See Appendix A, themes that emerged from participant narratives).

The final themes that emerged came from analyzing the different codes identified in the different focus groups conducted. (See Appendix A. Themes, subthemes, and codes that emerged from focus groups). The identified codes, which were most often described by the participants were as follows: lack of energy and disruption of daily activities; pain that is variable, intense, and disabling; increased sensitivity/irritability with others; mood swings; menstruation perceived as sickness; feeling understood. In all focus groups, there was a clear predominance of the following codes: lack of energy and disruption of daily activities with 91 codes, and pain that is variable, intense, and disabling, with 71 codes.

### 3.1. Theme 1. Living with Dysmenorrhea

The participants described what it is like to live with dysmenorrhea, based on the underlying experience of painful menstruation.

#### 3.1.1. Menstruation

Menstruation was perceived as “a disturbance or an illness”. Participants narrated how the information conveyed by their grandmothers and mothers regarding menstruation was full of myths and stereotypes. Thus, bloodstains on clothes were experienced with concern and shame, since menstruation is not openly discussed; rather, it is something that is covered up.

Menstruation was experienced negatively, as it produces changes and alters the normal functioning of women’s lives. It was defined as something awful, a sort of “living hell”, experienced with a sense of doom. This pain was sometimes accompanied by fear, uncertainty, along with great fatigue.

Some participants saw themselves as being ill. They recalled how their mothers and grandmothers referred to menstruation as a time of “sickness”. Participants acknowledged that menstruation has a negative connotation for their own health. The participants believed that this may stem from sociocultural traditions and from a transmission of myths and stereotypes among women, passed down from mother to daughter.

Menstruation creates a lot of uncertainty, both in terms of when it occurs and the amount of bleeding. This concern about leakage of blood appears while attending classes at the university and during clinical practice at the hospital. The possibility of blood leaking through to the white scrubs was a constant concern for the students in this study and a reason for distress and embarrassment. This feeling was accentuated during the night, while sleeping, because the participants were unable to avoid menstrual leaks. Thus, during the night, participants recounted that they have to be more vigilant about the bleeding, which in turn leads to difficulty sleeping and resting.

Moreover, menstruation is not dealt with openly: it is hidden, becoming a taboo. Situations such as asking for a tampon or a pad are done in a simulated way, hiding it, trying to avoid attracting attention.

#### 3.1.2. Pain

The pain of dysmenorrhea has no fixed pattern. It is changeable, both regarding the time of onset and in the intensity of the pain. Most participants described how the pain was more severe, intense, and disabling during the first few days of menstruation. However, this pain was variable and unpredic, appearing at different times, sometimes present throughout the entire menstruation, appearing a couple of days beforehand, increasing in intensity when menstruation is delayed, until it disappears and/or appears with great fluctuations in intensity, producing very intense acute pain, while for others, the pain was less intense, albeit more prolonged. Dysmenorrhea was accentuated in the presence of stress. In addition, not knowing when and with what intensity the pain will appear, and how it may limit their activities, made participants feel even more afraid, uncertain, and stressed.

Along with the pain, other symptoms may appear that worsen the experience of pain, such as dizziness, vomiting, diarrhea (or constipation), anxiety, fatigue, tiredness, swelling, heaviness, laziness, altered appetite, muscle and joint pain, migraines, headache, and altered sensation.

### 3.2. Theme 2. Body Changes and Mood Swings

During menstruation, body changes occur, which impact women’s perception of beauty, the clothes they wear, and their mood. The participants described how they “look uglier”, “deformed”, and felt unattractive due to the changes that menstruation produces in their bodies, feeling more swollen, especially in the abdomen, and with more acne. These changes caused them to adapt their clothing, as they felt uglier and because they needed comfortable clothes to avoid compressing certain parts of their body such as the abdomen and legs. The abdominal region was a particularly sensitive region for participants, as they perceived that menstruation causes them to swell, especially at the level of the abdomen, gaining volume in that region and increasing their overall body weight.

Participants also reported changes in their moods, experiencing emotional ups and downs, at times demanding greater gestures of affection or feeling discouraged, irritable, or seeking confrontation with people close to them. Some of the participants described this in their own language as feeling “bipolar”. These mood swings can also occur before menstruation. In addition to feeling discouraged, participants described a lack of energy, a lack of appetite, and decreased ability to carry out daily activities (study, work, etc.), grooming, and hobbies, sometimes having to interrupt these. The participants related how they perceived themselves as “more sensitive” in their relationship and behavior with other people. Thus, they were more receptive to displays of affection, patience, and care, although they were also more irritable when faced with behavior they did not like, criticism, or comments. This state of greater sensitivity sometimes made them feel guilty because they may have responded inappropriately to comments or observations from other people. The participants perceived this “increased sensitivity” and made an effort to control themselves. They identified hormonal changes during menstruation as the cause of this “sensitivity”, although some of them preferred not to acknowledge this, in order to not have to justify their behavior as a consequence of having their period: to avoid being judged.

### 3.3. Theme 3. Seeking a Safe Environment

Participants emphasized the need to have a safe environment, where they felt understood and could talk openly about dysmenorrhea, its pain, and sensations. Likewise, they identified environments considered unsafe, where they found incomprehension, and where the impact of dysmenorrhea was underestimated or neglected.

This safe environment was characterized by the following factors: (a) being a place where learners felt understood, “their greater sensitivity” was understood; (b) feeling understood when they were surrounded by women in similar circumstances, who had undergone or were experiencing the same pain, or with whom they have the confidence to talk; (c) being in places where they can talk freely about dysmenorrhea, and they perceive empathy for their situation; (d) when people close to them objectively saw the consequences of the pain in the participants, undergoing it with them and understanding the experience of living with dysmenorrhea.

On the contrary, the unsafe environment was characterized by the following factors: (a) people making comments, belittling the pain of the participants and playing down its importance, or judging them for not looking for an excuse not to work; (b) women who have not experienced pain during their menstruation, who are less understanding until they experience it in the first person; (c) women who have not had their menstruation for a long time and do not understand the participants, affirming that the pain is not so bad; (d) unfamiliar places or environments, where the participants do not have confidence to share their pain, are concerned about what is thought of them, and prefer to minimize their exposure.

According to the participants with dysmenorrhea, there are nuances in relation to whether an environment is safe or not. Being a person close to the participants such as family or a partner does not automatically determine that they belong to their safe environment. The same occurs with the expected negative attitude toward dysmenorrhea in men. While some participants stated that men did not understand their pain and despised them, other women narrated how they had not had any problems or felt despised, and that they had understood them when they told them of their pain. Belonging to the safe environment does not depend on gender but rather on a lack of empathy, which can also occur among women. Even within an internship and/or work environment, the participants described that it is the position, i.e., the person in charge (boss) who determines whether to communicate the reason of an absence due to dysmenorrhea, and not the person’s gender.

When in an unsafe environment, participants describe how they often hid the pain or justified absenteeism or a lack of productivity for reasons other than dysmenorrhea. The main reason for this was the fear that their superiors may believe that it is not worth hiring them because they will have frequent pain episodes and absenteeism. Criticism and lack of understanding in the workplace and/or clinical practice environment (in an environment considered unsafe) could be solved by communicating the real reasons for their pain. However, the participants described that they avoided doing this because they felt that it will harm them as women, as it may be used to underestimate or belittle them. This is paradoxical, because the participants described relationships among support, knowledge, and trust. The participants usually felt supported in the measure that their environment became aware of their pain; however, for this to happen, there must first be a relationship of trust where this information is disclosed.

## 4. Discussion

This study aimed to describe how nursing students experience body and mood changes due to PD. Our findings revealed that PD is a complex entity influenced by socio-cultural factors. For most women, menstruation has a negative connotation, and they tend to conceal their period. They experience great concern and uncertainty, as the course of their pain is unpredictable in terms of its duration, intensity, symptoms, and limitations. In addition, they perceive physical and psychological changes that make them feel less attractive and less willing to do activities and express themselves. In spite of all the above, women only talk freely about their problem when they perceive an environment as safe.

The participants described social and cultural influences based on family tradition in relation to myths and taboos about menstruation. This was reflected in the narratives, where participants described their periods as “the days I am sick” and the fact that women concealed their period. The negative attitude toward menstruation is present in all cultures and is still reflected in popular culture through myths and taboos [17,27,51]. Orringer et al. [52] noted how cultural factors in multi-ethnic groups play a central role in the meaning adolescents give to menstruation, especially in how information about menstruation and related issues is conveyed. Along these lines, previous studies [53,54,55] have identified deep-rooted social norms and cultural rules that lead to the concealment of menstruation, social pressure, and ongoing stress among women trying to comply with established rules. This also has an impact on women’s sexual health by interfering with health-care seeking and making it difficult to identify problems.

Current advertising campaigns for menstrual products also convey the constant need to conceal menstruation [17]. On the other hand, it is more common for several generations of women to suffer from PD in the same family, as different studies [5,56,57] have shown that a first-degree relative who has PD constitutes a risk factor for PD, and therefore, participants who have grown up in family settings where more women have suffered from menstrual pain would expect the association of menstruation–pain–sickness because of the limitations it entails. This is consistent with the “conceptual integrative model of transmission of risk from parents with chronic pain to offspring” proposed by Stone and Wilson [58]. In this model, the authors point out that the presence of pain inherently derives from the family and intergenerational ties, and it is based on genetic factors, specific social learning about pain, parenting and education, family health management, and exposure to a stressful environment. This model shows how it would be possible to create and target preventive interventions in subsequent generations to avoid or minimize pain.

The taboos and myths can result in gender inequality, which is manifested by avoiding or delaying consultation with health professionals for menstrual reasons, feelings of shame about menstruation, and the acceptance of the presence of pain as something inherent to womanhood [5,23]. In light of this, Wilson et al. [18] claim that it is necessary to dismantle menstrual taboos in order to overcome gender inequalities through early education interventions. However, Rastogi et al., in their systematic review of educational interventions to improve menstrual health, have highlighted that most of the current interventions do not produce significant changes in relation to these acquired taboos [59]. Other authors such as Yagnik are already working in this line, theorizing about new information models to mitigate these taboos [60]; however, it would also be interesting to further explore whether attitudes toward menstruation have any influence on the development or intensity of dysmenorrhea. In addition, it is essential that health professionals take into account cultural aspects and mythologies surrounding menstruation in order to provide care with a biocultural approach [27]. Previous studies such as that of Tan et al. [17] highlight how women describe their experience as an “all-round complex entity” despite the fact that the assessment of women with dysmenorrhea and the main lines of therapy are focused solely on pain and do not tend to meet their needs, as these are very standardized [12,61]. Therefore, this addresses the need for more comprehensive care. Burbeck et al. [27], in a phenomenological study of women in London, identified that their participants understood the menstrual process holistically and described multiple symptoms and aspects related to menstruation, rather than focusing on pain, presenting a more complex experience [27]. These results are consistent with previous studies [2,17,52] highlighting how women with PD tend to have higher levels of prostaglandins and greater sensitivity to pain, which contributes to a greater perception of pain in different locations and a more extensive experience of dysmenorrhea. 

In addition, women with dysmenorrhea strongly highlighted mood-related changes during their menstruation, such as increased sensitivity and susceptibility and even attempts at self-control that were shared by most participants and which are poorly described in previous studies. The women themselves attributed these to hormonal changes; however, as a novel finding, it is noted that they preferred to hide it, even making efforts toward greater self-control for fear of being judged by others. This is probably a consequence of the fear of being stigmatized by aspects related to menstruation, as Johnston-Robledo and Chrisler already pointed out [62]. They also perceived less appetite and lack of energy for performing activities on menstrual days. Along the lines of these findings, some previous quantitative studies [2,5,57] have identified greater irritability, depressive feelings, poorer sleep quality, and greater fatigue in women with dysmenorrhea during menstruation compared to women without dysmenorrhea. They also reported noticeable body changes that made them feel worse about themselves and that even conditioned their choice of clothing on those days. Krohmer et al. [63] identified variations throughout the menstrual cycle in terms of body satisfaction and selective attention toward different parts of the body. Both the findings of Krohmer et al. [63] as well as those in the present study could be attributed to hormonal changes; however, it is important to closely consider these findings, as body dissatisfaction is a risk factor for eating disorders. Studies need to be conducted on the experience and impact of body changes resulting from hormone levels typical of this phase of the cycle, and to identify whether there is any distortion of self-image in women suffering from dysmenorrhea compared to other women. 

Chrisler et al. [51] identified an association between positive attitudes toward menstruation and a more positive body image; however, our results give a worse assessment of body image during menstruation. This could be explained by the negative attitudes toward menstruation that they expressed for family reasons. 

The participants have narrated that stress aggravates the intensity of their pain. These results are consistent with previous studies [64,65,66] that identify stress as a potential risk factor for primary dysmenorrhea and attribute it to maladaptive responses that impact on cortisol secretion. The authors of the present study believe that a psychological assessment should be made before choosing the therapeutic strategy in patients with dysmenorrhea; as such, psychotherapy interventions could be an additional therapeutic option in some women [67].

Our results show how the presence of perceived support, knowledge of the problem, and a climate of trust helps to generate a safe environment where women can freely express aspects related to their menstrual pain. However, when they women detect potentially hostile environments, they react by hiding their symptoms for fear of criticism and lack of understanding. This is probably related to aspects identified by other authors such as the social normalization of menstrual pain referred to by Chen et al. [25,44] and the taboos that exist in society regarding menstruation and dysmenorrhea [33]. It is noteworthy that some women felt that expressing themselves in unsafe environments could be harmful to them as women, because it may be used to underestimate and belittle them. In this vein, Seear [68], in a study conducted among Australian women, noted that disclosure of menstrual issues can make women more vulnerable to stigmatization and lead to concealment. This finding poses enormous challenges in terms of education, social and health promotion, and women’s sexuality.

Previous studies on the presence of dysmenorrhea in groups of women of different professions show some similarities with our results. Thus, Kordi et al. [69], in their study on dysmenorrhea in Iranian midwives, describe an association between occupational stress and dysmenorrhea. In addition, László et al. [70] in a national survey of the Hungarian population identified an association between having dysmenorrhea and low levels of social support from co-workers and low job security. In contrast, the study by Chung et al. [71], among Taiwanese nurses, failed to find any association between dysmenorrhea and job satisfaction or stress. Therefore, it would be interesting to continue exploring this issue among health professionals from different professions and territories. 

Finally, the main strength of this study is the description of how young women live with dysmenorrhea, presenting first-time accounts that can influence patient care. To the best of our knowledge, this is one of the few studies that describes what it is like for young female students to live with PD.

This study has several limitations. Firstly, the results cannot be extrapolated; however, they can be applied to contexts with similar characteristics. This has been controlled via the exhaustive description of the study setting and methods, which enables other researchers to apply the same methods in other contexts (i.e., ensuring transferability) [26,27]. Secondly, the use of several data collection tools such as interviews may enrich the results obtained. In this study, participants did not agree to conduct individual interviews. In addition, within the FGs, the participants may tend to unify their criteria and avoid comparison and dialogue, which may produce uniform results [37,40]. To control for this effect, the moderator asked the participants questions to determine their individual perspective and to counteract this tendency. Finally, by conducting focus groups via the Internet through a digital platform, this could have reduced the interaction between participants and researchers. To avoid this, the study authors applied a detailed data collection procedure, using a rotational system to raise one’s hand and speak (see the FG procedure), and the number of participants in each group was reduced to minimize the overlap of interventions among participants. In addition, FGs carried out through an online platform (with the possibility of turning off the camera) would be a detriment to observer data. The reason for using online FGs was due to the enormous difficulties in gathering the students due to the demands of their personal and academic schedules. Nonetheless, none of the participants in the FGs turned off their camera.

## 5. Conclusions

This group of nursing students with PD considered menstruation to be negative and limiting, causing physical and mental changes, making women feel less attractive, and conditioning their way of dressing and relating to people in their environment. The main novelty of this study is the identification of safe and unsafe spaces that can condition the students’ experience of PD, who may improve thanks to social awareness strategies and improved policies. It is striking to consider that although these are future health professionals, we have identified a tendency toward concealment and a negative connotation of menstruation as aspects that can influence the attention given by health professionals, reflecting the need for specific training in this matter.

In addition, these results highlight the need for an assessment with a comprehensive biocultural approach that is focused on individuals and their needs and not only on the management and treatment of menstrual pain. Social and cultural elements are identified, such as the influence of the family in relation to the meaning of menstruation, together with fears of rejection and discrimination. These aspects pose a challenge for health professionals and especially for nurses and midwives to intervene not only for the treatment of pain, but also within the community, through educational and health promotion programs for women from childhood to adulthood. Among the practical implications, safe environments for student nurses should be promoted within clinical practice environments and at universities by implementing a variety of strategies. It is especially interesting that the results of further training and strategies may produce a positive impact beyond the students, since as future professionals, their experience can be reflected in the care they may later provide to other women and at the community level during their professional practice. These strategies could begin with the dissemination of evidence on the subject among teachers and clinical tutors and via measures such as offering students the possibility of changing their uniform at the centers should they stain their clothes, rethinking the need for white uniforms, and reviewing the rules of absenteeism and the academic impact of absences in relation to this problem.

## Figures and Tables

**Table 1 ijerph-17-06670-t001:** Data collection information.

	FG Session 1: Duration (minutes)	FG Session 2: Duration (minutes)	Participants	Age, Mean (SD)
FG1	51	52	6	21.5 (1.04)
FG2	65	46	6	22 (1.89)
FG3	51	47	6	24 (2.60)
FG4	45	45	6	24 (6.38)
FG5	78	42	9	22.3 (3.35)
**Total Number of FGs**	**Duration (minutes), Mean (SD), Session 1**	**Duration (minutes), Mean (SD), Session 2**	**Total Participants**	**Total Mean Age, Mean (SD)**
10	290.58 (13.37)	232, 46.4 (3.64)	33	22.72 (3.46)

FG: Focus Group.

**Table 2 ijerph-17-06670-t002:** Focus groups structure.

Phase	Contents	Time (min)
Moderator welcome	Welcome. Explanation of study aims, process of the session, and rules.	5–10
Opening question	Participants were asked about their experience with dysmenorrhea.	10–20
Introductory and transition questions	The question was centered on aspects of dysmenorrhea: body and mood changes.	10–30
Key questions	Questions were posed once more on the basis of prior participant responses in order to go into greater depth regarding areas such as living with dysmenorrhea.	20–40
Closing remarks	The moderator performed a brief summary of the contents covered.	10–15

**Table 3 ijerph-17-06670-t003:** Semi-structured focus group question guide.

Research Area	Questions
Living with dysmenorrhea	What is it like for you to live with dysmenorrhea? What is most relevant to you about this pain? How would you define or explain your pain?
Changes caused by dysmenorrhea	Have you experienced any changes during dysmenorrhea? Which of these changes are most relevant to you?
Impact of pain	How does this pain affect you? What is most relevant regarding its impact, living with pain? Does the pain have any impact on work or academic/study repercussions? In what way? What is most relevant about the pain during work/studies?
Understanding on behalf of family and friends	Has the pain influenced your relationships with other people? What about the relationship with your partner and/or the rest of the family? In what way?

**Table 4 ijerph-17-06670-t004:** Quality criteria.

Criteria	Techniques Performed and Application Procedures
Credibility	Investigator triangulation: each data source was analyzed. Thereafter, team meetings were performed during which the analyses were compared and themes were identified. Participant validation: This consisted of asking the participants to confirm the data obtained during the data collection stages.
Transferability	In-depth descriptions of the study performed, providing details of the characteristics of researchers, participants, contexts, sampling strategies, and the data collection and analysis procedures.
Dependability	Audit by an external researcher: An external researcher assessed the study research protocol, focusing on aspects concerning the methods applied and the study design.
Confirmability	Investigator triangulation and data collection triangulation. Researcher reflexivity was encouraged via the previous positioning, performance of reflexive reports, and by describing the rationale behind the study.

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
