# Peer review of "Living with Pain and Looking for a Safe Environment: A Qualitative Study among Nursing Students with Dysmenorrhea"

_ijerph, 2020, doi:10.3390/ijerph17186670_

Round 1

Reviewer 1 Report

The manuscript titled “Living with Pain and Looking for a Safe Environment. A Qualitative Study Using Focus Groups Among Nursing Students with Dysmenorrheais an interesting and important study. The design and approach of the study are adequate. In this study authors observed that focus groups of nursing students with Primary dysmenorrhea (PD) considered menstruation to be negative and limiting, causing physical and mood changes, making them feel less attractive, conditioning their way of dressing, and relating to people in their environment. I recommend this work for publication with minor revisions.

Minor comments;

  1. It is not clear what is the most common and uncommon consequence of Primary dysmenorrhea in the focus groups of nursing students.
  2. The authors should present the effect of PD in all 5 FGs using a table.
  3. Authors should also discuss whether women with PD in the different professions have a similar opinions about menstruation.

Author Response

The manuscript titled “Living with Pain and Looking for a Safe Environment. A Qualitative Study Using Focus Groups Among Nursing Students with Dysmenorrhea” is an interesting and important study. The design and approach of the study are adequate. In this study authors observed that focus groups of nursing students with Primary dysmenorrhea (PD) considered menstruation to be negative and limiting, causing physical and mood changes, making them feel less attractive, conditioning their way of dressing, and relating to people in their environment. I recommend this work for publication with minor revisions.

Minor comments;

  1. It is not clear what is the most common and uncommon consequence of Primary dysmenorrhea in the focus groups of nursing students.

Response: We do not fully understand what the reviewer means when he or she asks about the more or less common consequences of primary dysmenorrhea. The results obtained come from the thematic analysis of all the focus groups. They are based on the participants' experience with primary dysmenorrhea. This means, that they are the results obtained based on what they consider to be the most relevant.

Therefore, we understand that the reviewer asks about the presence of the codes and themes identified in the focus groups.

The identified codes, most narrated by the participants were (from highest to lowest): 1. Lack of energy and disruption of daily activities; 2. Pain that is variable, intense and disabling, 3. Increased sensitivity/irritability with others; 4. Mood swings; 5. Menstruation perceived as sickness; 6. Feeling understood. A clear predominance in all the focus groups was found for the following codes: Lack of energy and disruption of daily activities with 91 códigos and Pain that is variable, intense and disabling, with 71 codes.

Therefore, and related to the next question, the authors have included the following in the results section:

The final themes that emerged came from analyzing the different codes identified in the different focus groups conducted. (See Supplementary Materials Table S2. Themes, subthemes, and codes that emerged from Focus Groups). The identified codes, which were most often described by the participants were: Lack of energy and disruption of daily activities; Pain that is variable, intense and disabling; Increased sensitivity/irritability with others; Mood swings; Menstruation perceived as sickness; Feeling understood. In all focus groups there was a clear predominance of the following codes: Lack of energy and disruption of daily activities with 91 codes, and Pain that is variable, intense and disabling, with 71 codes.

Also, we have included a new Supplementary Materials Table S2. Themes, subthemes, and codes that emerged from Focus Groups. This new table shows the count of the themes and codes identified in each focus group. In this way one can observe the most common codes and themes obtained after the analysis.

  1. The authors should present the effect of PD in all 5 FGs using a table.

Response: A new table has been added (Supplementary Materials Table S2. Themes, subthemes, and codes that emerged from Focus Groups), where the counts of the codes obtained from each theme are described for each focus group. Thus, it is shown where the codes were identified and where they appeared for each focal group. Thus, some codes only appeared in some focal groups while other codes appeared in more than one focus group.

We included the following in the results sections:

The final themes that emerged came from analyzing the different codes identified in the different focus groups conducted. (See Supplementary Materials Table S2. Themes, subthemes, and codes that emerged from Focus Groups). The identified codes, which were most often described by the participants were: Lack of energy and disruption of daily activities; Pain that is variable, intense and disabling; Increased sensitivity/irritability with others; Mood swings; Menstruation perceived as sickness; Feeling understood. In all focus groups there was a clear predominance of the following codes: Lack of energy and disruption of daily activities with 91 codes, and Pain that is variable, intense and disabling, with 71 codes.

  1. Authors should also discuss whether women with PD in the different professions have a similar opinions about menstruation.

Response: We have followed the reviewer's recommendations. We have now included additional information in the discussion section.

Previous studies on the presence of dysmenorrhea in groups of women of different professions show some similarities with our results. Thus, Kordi et al. [70], in their study on dysmenorrhea in Iranian midwives, describe an association between occupational stress and dysmenorrhea. In addition, László et al. [71] in a national survey of the Hungarian population identified an association between having dysmenorrhea and low levels of social support from co-workers and low job security. In contrast, the study by Chung et al. [72], among Taiwanese nurses, failed to find any association between dysmenorrhea and job satisfaction or stress. Therefore, it would be interesting to continue exploring this issue among health professionals from different professions and territories.

We have included new references:

  1. Kordi, M.; Mohamadirizi, S.; Shakeri, M.T. The relationship between occupational stress and dysmenorrhea in midwives employed at public and private hospitals and health care centers in Iran (Mashhad) in the years 2010 and 2011. Iran. J. Nurs. Midwifery Res. 2013, 18, 316–22.
  2. László, K.D.; Gyorffy, Z.; Ádám, S.; Csoboth, C.; Kopp, M.S. Work-related stress factors and menstrual pain: A nation-wide representative survey. J. Psychosom. Obstet. Gynecol. 2008, 29, 133–138.
  3. Chung, F.; Yao, C.C.; Wan, G. The Associations between Menstrual Function and Life Style/Working Conditions among Nurses in Taiwan. J. Occup. Health 2005, 47, 149–156.

Reviewer 2 Report

This is an interesting piece of work about how women experience the process of primary dysmenorrhea using a sample of nursing students. Title: There is no need to add the data collection method (focus groups). Background: The authors describe the types of dysmenorrhea, symptomatology and treatments, but epidemiological data, worldwide prevalence, prevalence in Spain and prevalence among university students are not included. Cultural aspects linked to the problem should be addressed in this section. The inclusion criteria should be clearer--what exactly is meant by “young students”? Justify the reason for studying this population. If a gender perspective is used, what other perspectives have been used for this same objective? A theoretical framework is missing. The objective is well thought-out. Method: Define the design of the study more clearly--is it a phenomenological study? Is it descriptive phenomenology or hermeneutic? While it is valid to use focus groups, intimate information is not usually expressed in a group setting, so did the authors consider conducting in-depth interviews? The theoretical stance of the research team, and previous publications, show a broad pre-understanding of the issue. Gadamer’s phenomenology would have been a good theoretical-philosophical framework for this study. The focus groups carried out through an online platform (with the possibility of turning off the camera) would be a detriment to observer data, therefore why was this done in this way? What do you mean by focus groups having a uniform structure? Convenience sampling is indicated, however multiple studies emerge in student samples, so how did the authors ensure voluntary participation? Did participation form part of any evaluation or activity in any subject? Results: Well-developed. The authors are less daring in the inductive process of topic/sub-theme selection. For example: "feeling bipolar" or "finding confrontation," "sensitive and irritable" are more informative than "living with dysmenorrhea." Discussion: Well-developed as well, however it would have been improved if it was based on a theoretical model. It is interesting and well-written, although it is lengthy. In lines 321-326 the authors mention aspects such as changes in the intensity of pain, appearance, etc, which have been discussed in other studies. Any cultural aspects are scarcely discussed in general. Conclusions: The conclusions are well-presented, and respond well to the objectives. Nonetheless, reading the first paragraph, one might wonder, “What does this study add to what is already known about the problem?” The authors are encouraged to highlight the novelty of their results. It is also missing more concrete practical implications focused on nursing students, which make up the sample (such as concerns about spotting on white uniforms).

Author Response

This is an interesting piece of work about how women experience the process of primary dysmenorrhea using a sample of nursing students.

Title: There is no need to add the data collection method (focus groups).

Response: We have followed the reviewer's recommendation. We have modified the manuscript title.

Background: The authors describe the types of dysmenorrhea, symptomatology and treatments, but epidemiological data, worldwide prevalence, prevalence in Spain and prevalence among university students are not included.

Response: We have followed the reviewer's recommendations. We have included new information regarding dysmenorrhea within the introduction section.

Worldwide, the estimated prevalence is 66.6-75.2%. [1]. Concretely, in Spain, it is estimated to affect 56-62% of the female population [3,4], and around 75% of university students [5,6].

REFERENCES

  1. Armour, M.; Parry, K.; Manohar, N.; Holmes, K.; Ferfolja, T.; Curry, C.; MacMillan, F.; Smith, C.A. The Prevalence and Academic Impact of Dysmenorrhea in 21,573 Young Women: A Systematic Review and Meta-AnalysisJ. Women’s Health2019, 28, 1161–1171.
  2. Gómez-Escalonilla Lorenzo, B.; Rodríguez Guardia, Á.; Marroyo Gordo, J.M.; Mozas Lillo, R. de las Frecuencia y características de la dismenorrea en mujeres de la zona de salud de Torrijos (Toledo). Enfermería Clin.2010, 20, 32–35.
  3. Larroy, C.; Crespo, M.; Meseguer, C. Dismenorrea Funcional en la Comunidad Autónoma de Madrid: Estudio de la Prevalencia en Función de la Edad. Rev. Soc. Esp. Dolor.2001, 8, 11–22.
  4. Fernández-Martínez, E.; Onieva-Zafra, M.D.; Parra-Fernández, M.L. Lifestyle and prevalence of dysmenorrhea among Spanish female university students. PLoS ONE2018, 13, e0201894.
  5. Abreu-Sánchez, A.; Parra-Fernández, M.L.; Onieva-Zafra, M.D.; Ramos-Pichardo, J.D.; Fernández-Martínez, E. Type of Dysmenorrhea, Menstrual Characteristics and Symptoms in Nursing Students in Southern Spain. Healthcare 2020, 8, 302.

Cultural aspects linked to the problem should be addressed in this section.

Response: We have followed the reviewer's recommendations. We have included new information regarding cultural aspects and dysmenorrhea in the introduction section.

Previous studies have identified a great socio-cultural influence in relation to all aspects of menstruation, which has traditionally been a taboo subject around which there are numerous myths in different cultures [17,18]. These aspects must be taken into consideration given their potential influence on sexual health, since, for example, the concealment of menstrual aspects has a negative impact on the correct assessment and treatment of problems by health professionals [19].

REFERENCES

  1. Tan, D.A.; Haththotuwa, R.; Fraser, I.S. Cultural aspects and mythologies surrounding menstruation and abnormal uterine bleeding. Best Pract. Res. Clin. Obstet. Gynaecol. 2017, 40, 121–133.

  1. Wilson, E.; Haver, J.; Torondel, B.; Rubli, J.; Caruso, B.A. Dismatling menstrual taboos to overcome gender inequality. Lancet Child Adolesc. Health 2018, 2, e17.
  2. O’Flynn, N.; Britten, N. Diagnosing menstrual disorders: A qualitative study of the approach of primary care professionals. Br. J. Gen. Pract. 2004, 54, 353–358.

The inclusion criteria should be clearer--what exactly is meant by “young students”?

Response: The authors wrote in the abstract A qualitative exploratory study was performed among 33 young nursing students with PD.” The authors apologize, it was a mistake. The term "young" is not included in the criteria for inclusion in the main text. And this term is removed from the abstract. In Spain, all nursing students are over 18 years old.

Justify the reason for studying this population.

Response: We have followed the reviewer's recommendations. We have included new information in the introduction section.

At present, there are few international studies [25–27] and none in Spain, to describe and analyze the perspective and experience of young female students suffering from dysmenorrhea and how it influences their daily lives. Previous studies carried out in different countries agree that a large number of women of different ages normalize their condition and do not seek health care [14,28,29], which contrasts with other studies highlighting that dysmenorrhea negatively affects women’s quality of life, in the case of university students, it has an impact on academic performance and is related to absenteeism and presenteeism [1,23]. The majority of nursing students are young women, and therefore as such, they potentially experience menstrual pain. Absenteeism due to dysmenorrhea may be more of a concern for these students because their training requires a minimum of 2300 hours of clinical practice, according to European Directives, for the acquisition of their professional skills [30]. On the other hand, rigid regulations regarding attendance and control of absenteeism can increase presenteeism, understood as the practice of attending work or clinical training when a person is ill and not fully functioning [23,31]. Furthermore,  it has been shown that in the nursing context this practice increases the risk of medication errors and may decrease patient safety [32]. There is therefore a need to deepen the experience of nursing students with primary dysmenorrhea, which is likely to influence their perspective on caring for their future patients when they graduate.

REFERENCES

  1. Armour, M.; Parry, K.; Manohar, N.; Holmes, K.; Ferfolja, T.; Curry, C.; MacMillan, F.; Smith, C.A. The Prevalence and Academic Impact of Dysmenorrhea in 21,573 Young Women: A Systematic Review and Meta-Analysis. J. Women’s Heal. 2019, 28, 1161–1171.
  2. Fernández-Martínez, E.; Onieva-Zafra, M.D.; Abreu-Sánchez, A.; Fernández-Muñóz, J.J.; Parra-Fernández, M.L. Absenteeism during menstruation among nursing students in Spain. Int. J. Environ. Res. Public Health 2020, 17.
  3. Chen, C.X.; Shieh, C.; Draucker, C.B.; Carpenter, J.S. Reasons women do not seek health care for dysmenorrhea. J. Clin. Nurs. 2018, 27, 301–308.

  1. Parra-Fernández, M.L.; Onieva-Zafra, M.D.; Abreu-Sánchez, A.; Ramos-Pichardo, J.D.; Iglesias-López, M.T.; Fernández-Martínez, E. Management of Primary Dysmenorrhea among University Students in the South of Spain and Family Influence. Int. J. Environ. Res. Public Health 2020, 17, 5570.

  1. Directive 2013/55/EU of the European Parliament and Council of 20 November 2013 Amending Directive 2005/36/EC on the Recognition of Professional Qualifications and Regulation (EU) No 1024/2012 on Administrative Cooperation through the Internal Market Information System (‘the IMI Regulation’). Available online: https://www.boe.es/doue/2013/354/L00132-00170.pdf(accessed on 05 September 2020).
  2. Schoep, M.E.; Adang, E.M.M.; Maas, J.W.M.; De Bie, B.; Aarts, J.W.M.; Nieboer, T.E. Productivity loss due to menstruation-related symptoms: a nationwide cross-sectional survey among 32 748 women. BMJ Open 2019, 9, e026186.
  3. Critz, C.; Feagai, H.E.; Akeo, A.; Tanaka, M.; Shin, J.H.; Erickson, M.; Ikeda, M.; Moriya, H.; Ozaki, K. Sick Students: Presenteeism among Nursing Students in 3 Countries. Nurse Educ. 2020, 45, E1–E5.

If a gender perspective is used, what other perspectives have been used for this same objective? A theoretical framework is missing.

Response: We agree with reviewer. We used a constructivism framework. The authors believe that this theoretical framework helps us to understand people's perspectives on events and situations, and helps to plan interventions that are more focused on people's needs. The reality experienced by each person is constructed in relation to his or her environment, the people with whom he or she interacts, and the context.

In the present study a theoretical constructivist framework has been chosen, because it is a framework based on the meaning that people give to their experiences, in relation to the environment where they live, work and/or study. the meaning given to a phenomenon can vary, based on individuals, groups and institutions.

Carpenter & Suto (2008) regarding constructivism framework reported that: “…human beings construct their social reality, and that the social world cannot exist independently of human beings.” (p.23)

Creswell & Poth (2018) reported that: “In constructivism, individuals seek understanding of the world in which they live and work. They develop subjective meanings of their experiences-meanings directed toward certain objects or things. These meanings are varied and multiple, leading the researcher to look for the complexity of views (…). The goal of the research, then, is to rely as much as possible on the participants´views of the situation.”(p.24)

We have included in the introduction section:

The experience of these women needs to be described and analyzed to help implement care that addresses the perceived needs of women with dysmenorrhea, with the aim of providing comprehensive care from a gender perspective [22] and a constructivist framework [34]

We have included in the methods section:

Researchers based their approach on a a constructivist paradigm [39]. This paradigm was based on the assumption that human beings construct their own social reality, and that knowledge is built through increasingly nuanced reconstructions of individual or group experiences [37]. In constructivism, individuals develop meanings of their experiences. These meanings are varied and multiple, leading the researcher to seek a complexity of perspectives. The goal of the research is to rely on the participants´ views of the situation [39].

Reference:

  1. Carpenter C, Suto M. Qualitative Research for Occupational and Physical Therapist: A practical guide. Oxford, UK: Blackwell Publishing, 2008.
  2. Baxter, P.; Jack, S. Qualitative Case Study Methodology: Study Design and Implementation for Novice Researchers. The Qualitative Report. 2008, 13, 544–559.
  3. Creswell, J.W.; Poth, C.N. Qualitative Inquiry and Research Design. Choosing among Five Approaches, 4th ed.; Sage: Thousand Oaks, CA, USA, 2018.

The objective is well thought-out.

Response: Thank you for this comment.

Method: Define the design of the study more clearly--is it a phenomenological study? Is it descriptive phenomenology or hermeneutic? While it is valid to use focus groups, intimate information is not usually expressed in a group setting, so did the authors consider conducting in-depth interviews? The theoretical stance of the research team, and previous publications, show a broad pre-understanding of the issue. Gadamer’s phenomenology would have been a good theoretical-philosophical framework for this study.

Response:

In the present study A qualitative descriptive case study with a holistic single-case design was conducted.

The study of people's experiences in the face of diverse phenomena can be carried out through different types of qualitative designs, the most representative being phenomenology. However, there are other qualitative methods that also study these phenomena. This is the case of case studies. On this subject, Baxter and Jack (2008) reported the following regarding descriptive case studies …” This type of case study is used to describe an intervention or phenomenon and the real-life context in which it occurred.” (p.548).

The same authors (Carpenter & Suto, 2008.p.546) described an example of Case Example and The Research Questions:

Case Examples

The Research Questions

2. The experiences of 30-40 year old women following radical mastectomy faced with the decision of whether or not to undergo reconstructive surgery

How women (30-40 years of age) describe their post-op (first 6 months) experiences following a radical mastectomy? Do these experiences influence their decisions making related to breast reconstructive surgery?

Fàbregues & Fetters (2019) reported that: “Case study is a research design that involves an intensive and holistic examination of a contemporary phenomenon in a real-life setting (…) It uses to explore, describe or explain a single case bounded in time and place (ie, an event, individual, group, organisation or programme) (…) a case as ‘a phenomenon of some sort occurring in a bounded context”.

Also, case study designs: “…can be used to describe patients’ experiences regarding care, explore health professionals’ perceptions regarding a policy change, and understand why medical treatments and complex interventions succeed or fail.” (Fàbregues & Fetters, 2019.p.2)

And “Case study examines a specific phenomenon in detail.” (Fàbregues & Fetters, 2019.p.2)

Similarly, the qualitative design chosen (a case study), is consistent with the theoretical framework on which the study is based, constructivism. Phenomenology could not be used because it is based on a different theoretical framework, interpretationism. (Carpenter & Suto, 2008). The authors agree that Gadamer's phenomenology could have been one of the possible options for this study, as long as it had been based on an interpretative theoretical framework.

On the other hand, the data collection tool (focus groups), is consistent with the qualitative design chosen (case study). The case study design enables a wide range of instruments and data collection methods to investigate the phenomenon. Among these we find focus groups, observation and interviews.

Fàbregues & Fetters (2019) reported that: “Case study is often portrayed as a qualitative approach to research (eg, interviews, focus groups or observations)…”(p.6)

Moser & Korstjens (2018) reported that: “The purpose of the focus group discussion determines the composition. Smaller groups might be more suitable for complex (and sometimes controversial) topics. Also, smaller focus groups give the participants more time to voice their views and provide more detailed information, while participants in larger focus groups might generate greater variety of information (…) Focus groups generally consist of 6–12 participants.”(p.11)

In this sense, the authors were able to use the focus groups as the main data collection instrument. On the other hand, the participants refused to conduct personal interviews, therefore, the researchers had to use other data collection tools. This situation is included in the limitations section.

We have included in the limitations section:

Secondly, the use of several data collection tools such as interviews, may enrich the results obtained. In this study, participants did not agree to conduct individual interviews.

We have included in the design section:

A qualitative descriptive case study with a holistic single-case design was conducted [34,37,38] using focus groups (FGs). Qualitative methods are useful for understanding the beliefs, values, and motivations that underlie individual health behaviors [39]. A case study is a research design that examines a specific phenomenon in a real-life setting, and can be used to describe participants´ experiences regarding care or diseases [38]. A case study may be formed of different units, which help to describe a phenomenon. These units may be different participants who are connected by the phenomenon under study [34,37,38]. In this study, the phenomenon under study is the impact of PD among nursing students.

We have included this text in the data collection section:

In order to examine different perspectives within the same group, FGs were held to acquire an understanding of the problems faced by the group and to aid the identification of values and norms [30]. This method of data collection is congruent with the design of the case study [38]. Focus groups generally consist of 6–12 participants [43]. Also, smaller focus groups give the participants more time to voice their views and provide more detailed information, while participants in larger focus groups may generate greater variety of information [43].

REFERENCES:

  1. Carpenter C, Suto M. Qualitative research for occupational and physical therapists: A practical guide. Oxford: Black-Well Publishing; 2008
  2. Baxter P, Jack S. Qualitative Case Stuy Methodology: Study Design and Implementation for Novice Researchers. The Qualitative Report. 2008, 13, 544–559.
  3. Fàbregues, S.; Fetters, M.D. Fundamentals of case study research in family medicine and community health. Fam. Med. Community Health. 2019, 7, e000074.
  4. Moser, A.; Korstjens, I. Series: Practical guidance to qualitative research. Part 3: Sampling, data collection and analysis. Eur. J. Gen. Pract. 2018, 24, 9-18.

The focus groups carried out through an online platform (with the possibility of turning off the camera) would be a detriment to observer data, therefore why was this done in this way?

Response: We agree with the reviewer. The reason for using focus groups is because the participants did not agree to conduct individual interviews. On the other hand, within the chosen design (case study), the design enabled the use of a variety of instruments to collect the data (interviews, focus groups, observation) (Fàbregues & Fetters, 2019). Therefore, the authors used the focus groups as an instrument for data collection, as it was made possible by the study design, and when the interviews failed. In the original protocol both types of data collection instruments were foreseen however the authors could not apply one of these instruments. This degree of flexibility is accepted in qualitative methodology (Carpenter & Suto, 2008).

Afterwards, there were many difficulties and problems to conduct the focus groups in person, due to the students' schedules, which had theoretical classes at the University (from 5 to 8 pm) and clinical practices in hospitals (from 8 to 2 pm). The solution for performing the focus groups was to hold these online.

The authors believe that it is necessary to indicate this aspect in the limitations. We included the following in the limitations:

Also, FGs carried out through an online platform (with the possibility of turning off the camera) would be a detriment to observer data. The reason for using online FGs was due to the enormous difficulties in gathering the students, due to the demands of their personal and academic schedules. Nonetheless, none of the participants in the FGs turned off their camera.

References:

Carpenter C, Suto M. Qualitative research for occupational and physical therapists: A practical guide. Oxford: Black-Well Publishing; 2008

Fàbregues, S.; Fetters, M.D. Fundamentals of case study research in family medicine and community health. Fam. Med. Community Health. 2019, 7, e000074.

What do you mean by focus groups having a uniform structure?

Response: We meant that that the focus groups have a similar number of participants, ranging from 6 to 9 participants. The authors have made a mistake since we mistakenly used the word uniform in the wrong paragraph. The text has been modified and the term 'uniform structure’ has now been included in the correct paragraph.

We have modified the following paragraphs:

The FGs followed a uniform structure [48]. Each FG comprised between 6 and 9 participants, as group sizes below 4 can make it difficult to sustain a meaningful discussion and a group of >10 may prove difficult to manage [48].

The FGs were conducted by a moderator and an observer [50]. The moderator posed questions to which each participant responded, speaking in turns.

REFERENCES:

  1. Whalley Hammell, K.; Carpenter, C. Using qualitative focus groups to evaluate health programmes and service delivery. In Qualitative research in evidence-based rehabilitation; Whalley Hammell, K., Carpenter, C., Eds; London: Churchill Livingstone, 2004; 51–64.
  2. Bloor, M.; Frankland, J.; Thomas, M, Robson, K. Focus groups in social research. London: Sage, 2001

Convenience sampling is indicated, however multiple studies emerge in student samples, so how did the authors ensure voluntary participation? Did participation form part of any evaluation or activity in any subject?

Response: The researchers offered students the opportunity to participate in the study on a voluntary basis. To this end, the possibility of participating in the study was offered to all nursing students, enrolled in the University of Huelva (n= 403), in the first, second, third and fourth year of the nursing degree. Of those 403, 220 met the inclusion criteria and 43 agreed to participate voluntarily. In the present study, the inclusion of data collection was ceased at 33, due to redundancy of the information obtained (Moser & Korstjens (2018). Moreover, participation in the present study was not related to any academic activity or subject.

Moser & Korstjens (2018) reported: “ A guiding principle in qualitative research is to sample only until data saturation has been achieved. Data saturation means the collection of qualitative data to the point where a sense of closure is attained because new data yield redundant information. Data saturation is reached when no new analytical information arises anymore, and the study provides maximum information on the phenomenon (…) In quantitative research, by contrast, the sample size is determined by a power calculation.”(p.11)

We have included the following in the participants section:

The researchers offered all female nursing students from the four Degrees of Nursing at the University of Huelva the possibility of voluntarily participating. Participation in the study was unrelated to any subject or academic activity/work. Out of 403 students in total, 220 met the inclusion criteria, and 43 agreed to participate voluntarily.

REFERENCES:

Moser, A.; Korstjens, I. Series: Practical guidance to qualitative research. Part 3: Sampling, data collection and analysis. Eur. J. Gen. Pract. 2018, 24, 9-18.

Results: Well-developed.

Response: Thank you for this comment.

The authors are less daring in the inductive process of topic/sub-theme selection. For example: "feeling bipolar" or "finding confrontation," "sensitive and irritable" are more informative than "living with dysmenorrhea."

Response: The authors believe that the results presented are appropriate, since they are based on the participants' narratives. The authors have avoided inferring results that were not clearly based on the participants' narratives, or for which there were not enough codes. The authors identified the final topics based on the codes. In case of discrepancies, a final decision is made by consensus. The research group did not have enough information to be able to determine specific fields/areas for the terms pointed out by the reviewer.

Discussion: Well-developed as well, however it would have been improved if it was based on a theoretical model.

Response: In the present study, a previous theoretical model regarding previous DP care has not been used, because the authors, from their knowledge, have not found enough published literature about the experience of dysmenorrhea, which provides a complete theoretical model.

However, the authors have included in the discussion, the modelconceptual integrador of transmission of risk from parents with cronic pain to offspring” to discuss and relate findings regarding family influence on menstrual pain.

Current advertising campaigns for menstrual products also convey the constant need to conceal menstruation [17]. On the other hand, it is more common for several generations of women to suffer from PD in the same family, as different studies [5,57,58] have shown that a first-degree relative who has PD constitutes a risk factor for PD, and therefore, participants who have grown up in family settings where more women have suffered from menstrual pain would expect the association of menstruation-pain-sickness because of the limitations it entails. This is consistent with the "conceptual integrative model of transmission of risk from parents with chronic pain to offspring" proposed by Stone and Wilson [59].

NEW REFERENCE:

  1. Stone, A.L.; Wilson, A.C. Transmission of risk from parents with chronic pain to offspring: An integrative conceptual model. Pain. 2016, 157, 2628–2639.

It is interesting and well-written, although it is lengthy. In lines 321-326 the authors mention aspects such as changes in the intensity of pain, appearance, etc, which have been discussed in other studies.   

Response: We have followed the reviewer's recommendations. We removed the text from line 321 to 326.

“However, our results from this qualitative study also highlight relevant aspects not identified in the quantitative literature on symptomatology, such as the unpredictable clinical presentation in terms of the manifestation of symptoms, the intensity of the same, limitations due to the variability in relation to the onset of pain over different cycles, which generates great uncertainty and concern for women suffering from dysmenorrhea and limits them out of fear of its sudden appearance.”

On the other hand, it should be noted that other reviewers have asked to include more information in the discussion and therefore it has been difficult for us to reduce the length of the discussion.

Any cultural aspects are scarcely discussed in general.

Response: We have followed the reviewer's recommendations. We included new information regarding cultural aspects in the discussion section.

Orringer et al. [53] noted how cultural factors in multi-ethnic groups play a central role in the meaning adolescents give to menstruation, especially in how information about menstruation and related issues is conveyed. Along these lines, previous studies [54-56] have identified deep-rooted social norms and cultural rules that lead to concealment of menstruation, social pressure, and ongoing stress among women trying to comply with established rules. This also has an impact on women's sexual health, by interfering with health care seeking and making it difficult to identify problems.

NEW REFERENCES:

  1. Orringer, K.; Gahagan, S. Adolescent girls define menstruation: A multiethnic exploratory study. Health Care Women Int. 2010, 31, 831–847
  2. Brantelid, I.E.; Nilvér, H.; Alehagen, S. Menstruation During a Lifespan: A Qualitative Study of Women’s Experiences. Health Care Women Int. 2014, 35, 600–616.
  3. Uskul, A.K. Women’s menarche stories from a multicultural sample. Soc. Sci. Med. 2004, 59, 667–679.
  4. Çevirme, A.S.; Çevirme, H.; Karaoǧlu, L.; Uǧurlu, N.; Korkmaz, Y. The perception of menarche and menstruation among Turkish married women: Attitudes, experiences, and behaviors. Soc. Behav. Pers. 2010, 38, 381–394.

Conclusions: The conclusions are well-presented, and respond well to the objectives. Nonetheless, reading the first paragraph, one might wonder, “What does this study add to what is already known about the problem?” The authors are encouraged to highlight the novelty of their results. It is also missing more concrete practical implications focused on nursing students, which make up the sample (such as concerns about spotting on white uniforms).

Response: Thank you for these comments and for the positive feedback. Regarding the following comments, we have followed the reviewer's recommendations.

We have included the following in the conclusions section:

This group of nursing students with PD considered menstruation to be negative and limiting, causing physical and mental changes, making women feel less attractive, conditioning their way of dressing, and relating to people in their environment. The main novelty of this study is the identification of safe and unsafe spaces that can condition the students’ experience of PD, who may improve thanks to social awareness strategies and improved policies. It is striking to consider that although these are future health professionals, we have identified a tendency towards concealment and a negative connotation of menstruation, as aspects that can influence the attention given by health professionals, and reflecting the need for specific training in this matter.

Also, these results highlight the need for an assessment with a comprehensive biocultural approach, focused on individuals and their needs and not only on the management and treatment of menstrual pain. Social and cultural elements are identified, such as the influence of the family in relation to the meaning of menstruation, together with fears of rejection and discrimination. These aspects pose a challenge for health professionals and especially for nurses and midwives, to intervene not only for the treatment of pain, but also within the community, through educational and health promotion programs for women, from childhood to adulthood. Among the practical implications, safe environments for student nurses should be promoted within clinical practice environments and at universities by implementing a variety of strategies. It is especially interesting that the results of further training and strategies may produce a positive impact not only on the students, but also, as future professionals, their experience can be reflected in the care they may later provide to other women and at the community level during their professional practice. These strategies could begin with the dissemination of evidence on the subject among teachers and clinical tutors and via measures such as offering students the possibility of changing their uniform at the centers should they stain their clothes, rethinking the need for white uniforms and reviewing the rules of absenteeism and the academic impact of absences in relation to this problem.

Reviewer 3 Report

This manuscript focuses on the process experienced by nursing students in relation to dysmenorrhea. The rationale for this study is the importance of this disruption as a cause of absenteeism and physical and psychological disorders; as well as the paucity of studies of a qualitative nature in Spain.

In general, the manuscript is well organized and the results are relevant to increase the theoretical corpus of the discipline.

Some suggestions should be taken into consideration by the authors:

- The work is proposed with a general objective, it would be interesting to add specific objectives that give response to the obtained results

- In the section of material and method,

2.1 You must justify why is the technique of discussion groups used instead of other type of techniques, What advantages does this technique provide for this study?

  1. 2 The description of the research team does not provide information to the study. It is necessary to extend the concept of the constructivist paradigm and reason it for this work.

I would suggest to use a model or theory that would provide consistency to the results

- Line 97. There’s no need to define exclusion criteria, the possible participants of the study are well defined with the inclusion criteria.

- Line 162. Review Table . 4. Add C to credibility criteria

Author Response

This manuscript focuses on the process experienced by nursing students in relation to dysmenorrhea. The rationale for this study is the importance of this disruption as a cause of absenteeism and physical and psychological disorders; as well as the paucity of studies of a qualitative nature in Spain. In general, the manuscript is well organized and the results are relevant to increase the theoretical corpus of the discipline.

Response: Thank you for this comment.

Some suggestions should be taken into consideration by the authors:

- The work is proposed with a general objective, it would be interesting to add specific objectives that give response to the obtained results.

Response: We have followed the reviewer's recommendations.

Also, we have included the following in the introduction section:

The objectives of our study were to describe: a) how nursing students experienced PD, b) how students experienced the associated body and mood changes, and c) the impact of pain in their lives.

In the section of material and method,

2.1 You must justify why is the technique of discussion groups used instead of other type of techniques, What advantages does this technique provide for this study?

Response: The reviewer is right. One of the reasons for using focus groups is because the participants did not agree to conduct individual interviews. On the other hand, within the chosen design (case study), the design enabled the use of a variety of instruments to collect the data (interviews, focus groups, observation) (Fàbregues & Fetters, 2019). Therefore, the authors used the focus groups as an instrument for data collection, when the interviews failed. In addition, the focus groups enable the ability to share experiences among a group of people who have experienced the same event, situation or experience, allowing  to share and find meaning for that experience (Carpenter & Suto, 2008). The data collection tool (focus groups), is consistent with the qualitative design chosen (case study).

Fàbregues & Fetters (2019) reported that: “Case study is often portrayed as a qualitative approach to research (eg, interviews, focus groups or observations)…”(p.6)

These two circumstances, the fact that the Focus Groups are coherent with the chosen design (case study) and that the participants refused to conduct individual interviews, have been included within the manuscript.

We have included the following in the design section:

Also, FGs were used to study unexplored situations or those that were difficult to access via other methods [34,40], as well as confirming the hypothesis, developing questionnaires and designing intervention programs [40].

We have included the following in the data collection section:

In order to examine different perspectives within the same group, FGs were held to acquire an understanding of the problems faced by the group and to aid the identification of values and norms [30]. This method of data collection is congruent with the design of the case study [38]. Focus groups generally consist of 6–12 participants [43]. Also, smaller focus groups give the participants more time to voice their views and provide more detailed information, while participants in larger focus groups may generate greater variety of information [43].

We have included the following in the limitations section:

Secondly, the use of several data collection tools such as interviews, may enrich the results obtained. In this study, participants did not agree to conduct individual interviews.

REFERENCES:

  1. Carpenter C, Suto M. Qualitative research for occupational and physical therapists: A practical guide. Oxford: Black-Well Publishing; 2008
  2. Fàbregues, S.; Fetters, M.D. Fundamentals of case study research in family medicine and community health. Fam. Med. Community Health. 2019, 7, e000074.
  3. Barbour, R.S. Making sense of focus groups. Med. Educ.2005, 39, 742–750.
  4. Moser, A.; Korstjens, I. Series: Practical guidance to qualitative research. Part 3: Sampling, data collection and analysis. Eur. J. Gen. Pract. 2018, 24, 9-18.

2.2 The description of the research team does not provide information to the study. It is necessary to extend the concept of the constructivist paradigm and reason it for this work.

Response: The reason for including the research team section is because The guidelines for conducting qualitative studies established by the Consolidated Criteria for Reporting Qualitative Research (COREQ) (Tong et al., 2007) and the Standards for Reporting Qualitative Research (SRQR) (O´Brien et al., 2014) recommend including these sections as essential elements of the structure of a qualitative study.

O´Brien et al (2014) regarding Researcher characteristics and reflexivity section reported that: “Researchers’ characteristics that may influence the research, including personal attributes, qualifications/experience, relationship with participants, assumptions, and/or presuppositions; potential or actual interaction between researchers’ characteristics and the research questions, approach, methods, results, and/or transferability.”

On the other hand, the constructivist theoretical framework involves describing the positioning and characteristics of the research team.

In this way, Creswell & Poth (2018) reported that: “Thus, constructivist researchers often address the processes of interaction among individuals (…) Researchers recognize that their own background shapes their interpretation, and they “position themselves” in the research to acknowledge how their interpretation flows from their own personal, cultural ,and historical experiences.”(p.24)

In the present study, the theoretical constructivist framework has been chosen, because it is a framework based on the meaning that people give to their experiences, in relation to the environment where they live, work and/or study. That meaning given to a phenomenon can vary, based on individuals, groups and institutions.

Creswell & Poth (2018) reported that: “In constructivism, individuals seek understanding of the world in which they live and work. They develop subjective meanings of their experiences-meanings directed toward certain objects or things. These meanings are varied and multiple, leading the researcher to look for the complexity of views (…). The goal of the research, then, is to rely as much as possible on the participants´views of the situation.”(p.24)

On the other hand, we have increased the information about the paradigm or theoretical constructivist framework.

The following text has been included:

Researchers based their approach on a constructivist paradigm [39]. This paradigm was based on the assumption that human beings construct their own social reality, and that knowledge is built through increasingly nuanced reconstructions of individual or group experiences [37]. In constructivism, individuals develop meanings of their experiences. These meanings are varied and multiple, leading the researcher to seek a complexity of perspectives. The goal of the research is to rely on the participants´ views of the situation [39].

REFERENCES:

O'Brien BC, Harris IB, Beckman TJ, et al. Standards for reporting qualitative research: a synthesis of recommendations. Acad Med. 2014;89(9):1245-1251. doi: 10.1097/ACM.0000000000000388.

Creswell, J.W.; Poth, C.N. Qualitative Inquiry and Research Design. Choosing among Five Approaches, 4th ed.; Sage: Thousand Oaks, CA, USA, 2018.

Tong A, Sainsbury P, Craig J. Consolidated criteria for reporting qualitative research (COREQ): a 32-item checklist for interviews and focus groups. Int J Qual Health Care. 2007;19(6):349-357. doi: 10.1093/intqhc/mzm042.

  1. Baxter, P.; Jack, S. Qualitative Case Study Methodology: Study Design and Implementation for Novice Researchers. The Qualitative Report. 2008, 13, 544–559.

  1. Creswell, J.W.; Poth, C.N. Qualitative Inquiry and Research Design. Choosing among Five Approaches, 4th ed.; Sage: Thousand Oaks, CA, USA, 2018.

I would suggest to use a model or theory that would provide consistency to the results.

Response: In the present study, a previous theoretical model regarding DP care has not been used, because the authors, from their knowledge, have not found enough published literature about the experience of dysmenorrhea, to provide a complete theoretical model.

However, in the discussion, the authors include the "conceptual integrative model of transmission of risk from parents with chronic pain to offspring" to discuss and relate the results related to the family influence on menstrual pain.

“Current advertising campaigns for menstrual products also convey the constant need to conceal menstruation [17]. On the other hand, it is more common for several generations of women to suffer from PD in the same family, as different studies [5,57,58] have shown that a first-degree relative who has PD constitutes a risk factor for PD, and therefore, participants who have grown up in family settings where more women have suffered from menstrual pain would expect the association of menstruation-pain-sickness because of the limitations it entails. This is consistent with the "conceptual integrative model of transmission of risk from parents with chronic pain to offspring" proposed by Stone and Wilson [59]. In this model, the authors point out that the presence of pain inherently derives from the family and intergenerational ties, and is based on genetic factors, specific social learning about pain, parenting and education, family health management, and exposure to a stressful environment. This model shows how it would be possible to create and target preventive interventions in subsequent generations to avoid or minimize pain.

A NEW REFERENCE HAS BEEN INCLUDED:

  1. Stone, A.L.; Wilson, A.C. Transmission of risk from parents with chronic pain to offspring: An integrative conceptual model. Pain. 2016, 157, 2628–2639.

Line 97. There’s no need to define exclusion criteria, the possible participants of the study are well defined with the inclusion criteria.

Response: We have followed the reviewer's recommendations. We removed exclusion criteria.

Line 162. Review Table . 4. Add C to credibility criteria

 Response: We have followed the reviewer's recommendations.